# *SPOP* mutation leads to genomic instability in prostate cancer

Gunther Boysen[1,2,11†], Christopher E Barbieri[3,4*†], Davide Prandi[5], Mirjam Blattner[6], Sung-Suk Chae[6], Arun Dahija[6], Srilakshmi Nataraj[6], Dennis Huang[6], Clarisse Marotz[6], Limei Xu[6], Julie Huang[6], Paola Lecca[5], Sagar Chhangawala[7,8], Deli Liu[3,8], Pengbo Zhou[6], Andrea Sboner[6,8,10], Johann S de Bono[2,11], Francesca Demichelis[5,8,10‡], Yariv Houvras[7,9‡], Mark A Rubin[3,4,6,10*‡]

[1]Department of Pathology and Laboratory Medicine, Weill Cornell Medical College, New York, United States; [2]Division of Clinical Studies, Institute of Cancer Research, London, United Kingdom; [3]Department of Urology, Weill Cornell Medical College, New York, United States; [4]Sandra and Edward Meyer Cancer Center, Weill Cornell Medical College, New York, United States; [5]Centre for Integrative Biology, University of Trento, Trento, Italy; [6]Department of Pathology and Laboratory Medicine, Weill Cornell Medical College, New York, United States; [7]Department of Surgery, Weill Cornell Medical College, New York, United States; [8]HRH Prince Alwaleed Bin Talal Bin Abdulaziz Alsaud Institute for Computational Biomedicine, Weill Cornell Medical College, New York, United States; [9]Department of Medicine, Weill Cornell Medical College, New York, United States; [10]Institute for Precision Medicine, New York-Presbyterian Hospital, Weill Cornell Medical College, New York, United States; [11]The Royal Marsden, London, United Kingdom

*For correspondence: chb9074@
med.cornell.edu (CEB); rubinma@
med.cornell.edu (MAR)

†These authors contributed
equally to this work

‡These authors also contributed
equally to this work

Competing interests:
See page 15

Reviewing editor: Joaquín M
Espinosa, University of Colorado
at Boulder, United States

**Abstract** Genomic instability is a fundamental feature of human cancer often resulting from impaired genome maintenance. In prostate cancer, structural genomic rearrangements are a common mechanism driving tumorigenesis. However, somatic alterations predisposing to chromosomal rearrangements in prostate cancer remain largely undefined. Here, we show that *SPOP*, the most commonly mutated gene in primary prostate cancer modulates DNA double strand break (DSB) repair, and that *SPOP* mutation is associated with genomic instability. In vivo, *SPOP* mutation results in a transcriptional response consistent with *BRCA1* inactivation resulting in impaired homology-directed repair (HDR) of DSB. Furthermore, we found that SPOP mutation sensitizes to DNA damaging therapeutic agents such as PARP inhibitors. These results implicate SPOP as a novel participant in DSB repair, suggest that *SPOP* mutation drives prostate tumorigenesis in part through genomic instability, and indicate that mutant *SPOP* may increase response to DNA-damaging therapeutics.

## Introduction

Genomic instability is a fundamental feature of human cancer, and DNA repair defects resulting in impaired genome maintenance promote pathogenesis of many human cancers (*Hanahan and Weinberg, 2011*; *Garraway and Lander, 2013*). In prostate cancer, structural genomic rearrangements, including translocations (e.g., *TMPRSS2-ERG*) and copy number aberrations (e.g., 8q gain, 10q23/*PTEN* loss) are a key mechanism driving tumorigenesis (*Visakorpi et al., 1995*; *Cher et al., 1996*; *Tomlins et al., 2005*; *Zhao et al., 2005*; *Liu et al., 2006*; *Demichelis et al., 2009*; *Beroukhim et al., 2010*).

Whole genome sequencing (WGS) has allowed an unprecedented insight into the alterations underlying cancer. Recently, WGS of treatment naive, clinically localized prostate cancer revealed a striking

**eLife digest** Prostate cancer is the most common type of cancer in men in the UK and USA. Cancers develop when cells in the body acquire genetic mutations that allow the cells to grow rapidly and form a mass known as a tumor. Prostate cancer cells from different individuals can carry different genetic mutations, which affects whether the disease progresses and how the tumors respond to medical treatments. This genetic variety arises in cancer cells partly from a phenomenon known as genomic instability, in which DNA mutations accumulate due to defects in DNA repair.

Genetic studies of biopsies taken from human prostate cancers have shown that genomic instability causes chromosomes—the structures in which the cell's DNA is organized—to break and then be stuck back together haphazardly. As a result, fragments of chromosomes can end up in the wrong position, be duplicated, or be lost altogether. All of these mutations could spur on the growth of the tumor. However, it is currently not clear why some prostate cancers are more genomically unstable than others, or what exactly causes this instability.

Boysen, Barbieri et al. studied prostate cancer cells taken from patients before they started medical treatment. The experiments show that the cancer cells with high levels of genomic instability also often had mutations in a gene that encodes a protein called SPOP. These mutations occur in about 10 percent of men with prostate cancer and appear early in the development of the tumors. Next, they studied the SPOP protein in zebrafish (which is nearly identical to human SPOP), as well as in mouse and human cells. The experiments show that SPOP normally helps the cell to accurately repair DNA that has been damaged. Mutations in SPOP change the DNA repair process, which lead to genomic instability by increasing the likelihood that broken chromosomes will be stuck back together incorrectly.

Further experiments tested drugs known as PARP inhibitors on mouse and human prostate cancer cells. The drugs, which have been recently tested successfully in patients with prostate cancer, block a different method of DNA repair that operates separately to the one that involves SPOP. When both of these pathways were inactivated—one by the SPOP mutation, the other by the drug—the cancer cells died more quickly. Therefore, men that are diagnosed with types of prostate cancer in which the gene that encodes SPOP is mutated might benefit from treatment with PARP inhibitors or other therapies that affect DNA repair.

abundance of genomic rearrangements, in some samples comparable to the number of rearrangements in metastatic prostate cancer (*Baca et al., 2013*). Furthermore, the type (intrachromosomal vs interchromosomal) and complexity of rearrangements in these tumors shows remarkable heterogeneity, potentially suggesting distinct mechanisms of instability in different molecular classes of prostate cancer. However, somatic alterations underlying these phenomena remain unexplained.

Mutations in *SPOP* (Speckle-type POZ protein) occur in around 10% of prostate cancers and represent the most common non-synonymous mutations in primary prostate cancer (*Barbieri et al., 2012*). *SPOP* mutations define a distinct molecular class of prostate cancer; they are mutually exclusive with ETS rearrangements but display distinct patterns of somatic copy number alterations (SCNAs) (*Barbieri et al., 2012*).

Here, we investigated somatic alterations associated with genomic rearrangements in prostate cancer. We show that *SPOP* mutation is an early event specifically associated with increased intrachromosomal genomic rearrangements. Mechanistically, in vitro and in vivo data suggest that SPOP participates in repair of DNA double strand breaks (DSB), and *SPOP* mutation impairs homology-directed repair (HDR), instead promoting error-prone non-homologous end joining (NHEJ).

## Results

To nominate somatic events associated with structural genomic rearrangements in clinically localized prostate cancer, we examined WGS data from 55 treatment naive prostate cancers (*Baca et al., 2013*) (*Figure 1A*). This analysis revealed a bimodal distribution, with a more common, 'low-rearrangement' population, and a less frequent 'high-rearrangement' population primarily driven by intrachromosomal rearrangements (deletions, inversions, and tandem duplications), rather than balanced interchromosomal rearrangements (*Figure 1B*). We then analyzed the association between recurrent

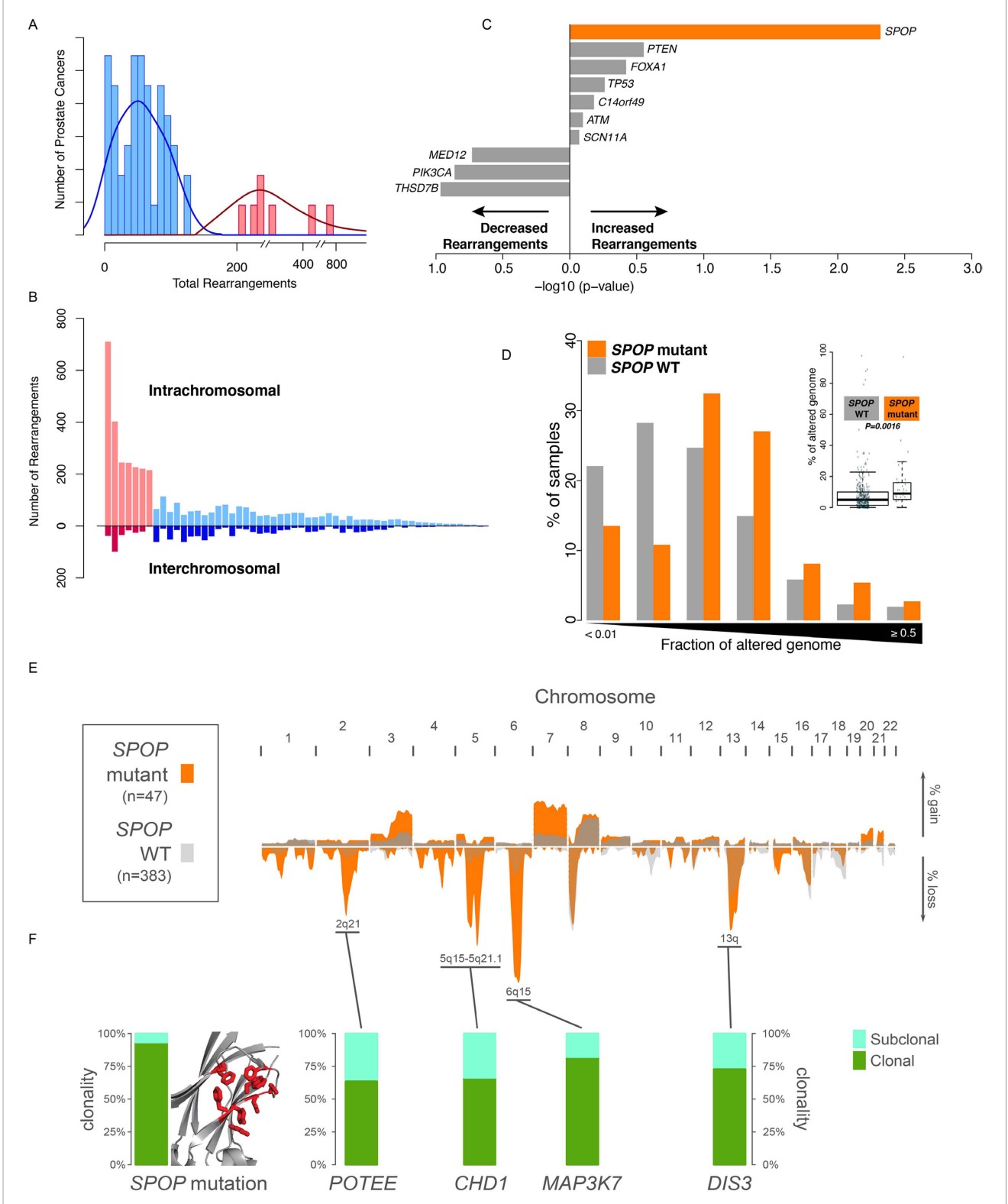

**Figure 1**. SPOP mutant prostate cancer displays increased genomic rearrangements. (**A**) Distribution analysis of genomic rearrangements from 55 clinically localized prostate cancers distinguishes two subpopulations. (**B**) Increased total rearrangements are driven by intrachromosomal rather than interchromosomal rearrangements. 55 clinically localized prostate cancers ordered (right to left) according to total rearrangements; numbers of intrachromosomal and interchromosomal rearrangements are displayed. (**C**) Association of recurrent point mutations with intrachromosomal

*Figure 1. continued on next page*

*Figure 1. Continued*

rearrangements. X-axis shows (−log10) p-value. (**D**) SPOP mutant prostate cancers harbor increased total somatic copy number aberration (SCNA) burden. The fraction of altered genome, partitioned into bins covering a range from <0.01 to ≥0.5, is shown as a histogram for SPOP WT and SPOP mutant tumors. Inset: the percentage of altered genome is significantly increased in SPOP mutant prostate cancers (p = 0.0016, two-sample Wilcoxon-Mann-Whitney test). (**E**) Frequency of somatic copy number alterations in 430 clinically localized prostate cancers. SPOP-mutant cancers (orange) and SPOP-wild-type tumors (gray). Length of bars indicates the frequency of copy number alterations. (**F**) Clonality of selected alterations associated with genomic rearrangements, in 430 clinically localized prostate cancers.

The following figure supplements are available for figure 1:

**Figure supplement 1**. Association of recurrent SCNAs with intrachromosomal rearrangements.

**Figure supplement 2**. Association of recurrent SCNAs or point mutations with interchromosomal rearrangements.

**Figure supplement 3**. Genomic view of CHD1 and MAP3K7 areas in 430 prostate cancers sorted by SPOP mutation status.

**Figure supplement 4**. Genomic burden of three subset of prostate cancers defined by the aberration state of SPOP, CHD1, and MAP3K7.

**Figure supplement 5**. Evolution graphs built from prostate cancer sequencing data.

somatic alterations (point mutations and SCNAs) and number of rearrangements (*Figure 1C*; *Figure 1—figure supplements 1, 2*). Several recurrent deletions, primarily on chromosomes 5q and 6q, were significantly associated with intrachromosomal genomic rearrangements (*Figure 1—figure supplement 1*), and these were completely distinct compared to alterations associated with interchromosomal rearrangements (*Figure 1—figure supplement 2*). Among recurrent point mutations, only a single lesion—mutation in *SPOP*—was significantly associated with increased rearrangements (*Figure 1C*). Consistent with increased intrachromosomal rearrangements, SCNA analysis revealed that *SPOP* mutant prostate cancers showed significantly higher total copy number alteration burden (*Figure 1D*).

*SPOP* mutation frequently co-occurs with specific SCNAs, designating a molecular class of prostate cancer (*Barbieri et al., 2012*; *Blattner et al., 2014*) (*Figure 1—figure supplement 3*). Independent analysis of SCNAs from three publicly available data sets comprising 430 tumors (*Baca et al., 2013*; *Barbieri et al., 2012*; *Consortium TCGA, 2015*), including 47 *SPOP* mutant prostate cancers, confirmed that the rearrangement-associated deletions (*Figure 1—figure supplement 1*) were those enriched in *SPOP* mutant prostate cancer (*Figure 1E*). When individually comparing *SPOP* mutations and associated deletions (*CHD1* and *MAP3K7*), we did not observe significant differences in SCNA burden for any one lesion (*Figure 1—figure supplement 4*). Analysis of clonality (*Baca et al., 2013*; *Prandi et al., 2014*) of specific lesions showed that *SPOP* mutations were highly clonal compared to loci in the associated deletion peaks, supporting that *SPOP* mutations precede deletions (*Figure 1F*). In addition, analysis of dependencies of the lesions supports *SPOP* mutations preceding *CHD1* deletions; no lesions were predicted to precede *SPOP* mutation (*Figure 1—figure supplement 5*). Together, these data nominate a distinct prostate cancer class characterized by early *SPOP* mutations and genomic instability. We posited that the *SPOP* mutation impacts genome maintenance and prioritized *SPOP* for functional studies.

We explored the functional role of SPOP in vivo, using zebrafish as a rapidly assessable vertebrate model system. SPOP is highly conserved (97.3% identical at the amino acid level between human and zebrafish, *Figure 2—figure supplement 1A*). Knockdown of *Spop* by two different splice-blocking morpholinos (MO5, MO7) dramatically impaired brain and eye development as well as decreased overall body size (*Figure 2A,B*; *Figure 2—figure supplement 1F*), resulted in gene expression changes consistent with p53 activation, and apoptosis measured by TUNEL assay (*Figure 2C*, *Figure 2—figure supplement 1E*). Microinjection of human SPOP mRNA rescued these phenotypes, confirming specificity of the morpholino effects (*Figure 2A–C*, *Figure 2—figure supplement 1F*). To nominate signaling pathways impacted by Spop, we performed transcriptional profiling using RNA-seq on zebrafish with *Spop* knockdown and ectopic expression of wild-type SPOP (SPOP-wt) and

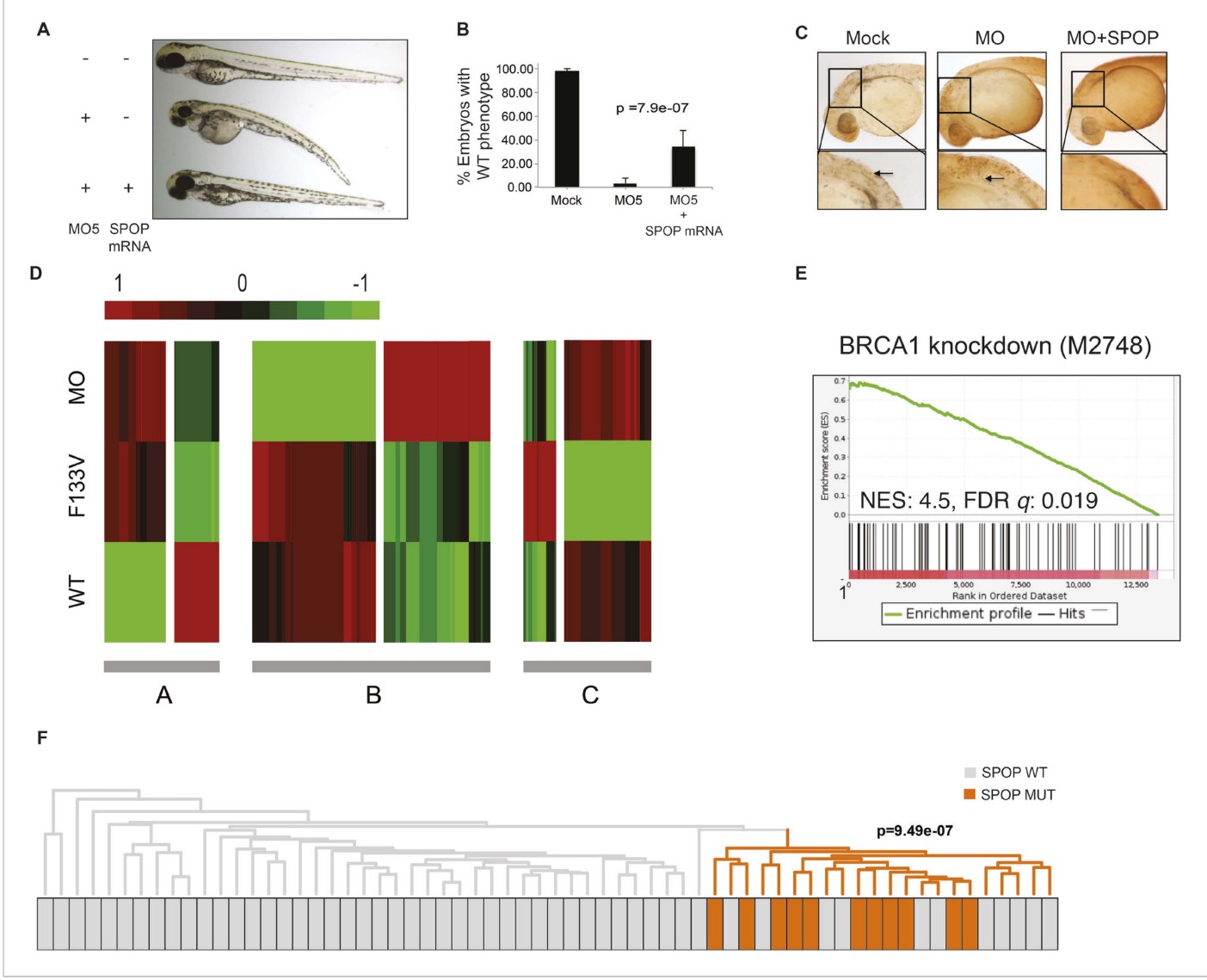

**Figure 2**. SPOP mediates DNA damage repair. (**A**) Evaluation of SPOP function during zebrafish development. Phenotype of morpholino-mediated *Spop* knockdown (MO5) in zebrafish embryos at 70 hr post fertilization (hpf). Injection of human SPOP mRNA (250 pg) rescued the phenotype. (**B**) Quantification of the rescue of SPOP phenotype after ectopic expression of human SPOP mRNA. Results are represented as s.e.m. (**C**) Whole mount TUNEL assay to determine apoptosis in zebrafish embryos. Arrows point to apoptotic cells (brown). Shown are representative images. (**D**) Heatmap representation of gene expression differences in zebrafish embryos ectopically expressing SPOP-wt or SPOP-F133V compared to SPOP knockdown by morpholino (MO). The list of genes can be found in *Figure 2—figure supplement 3*. Number of genes per block: A (198), B (429), C (223). (**E**) Gene set enrichment analysis (GSEA) of RNA sequencing data derived from zebrafish embryos expressing SPOP-wt or SPOP-F133V (24 hpf). Enrichment plot for the BRCA1 gene signature is shown. Molecular Signatures Database (MSigDB) systematic name indicated in brackets. (NES) Normalized Enrichment Score. (FDR) False Discovery Rate. (**F**) Dendrogram of human primary prostate cancer cases based on BRCA1 knockdown genes (MSigDB: M2748). Unsupervised clustering of RNA-seq data from human primary prostate cancer with wild-type (n = 53) or mutant SPOP (n = 11), performed on the BRCA1 knockdown gene signature (M2748) identified in zebrafish embryos by GSEA as shown in (**E**).

The following source data and figure supplements are available for figure 2:

**Source data 1**. List of genes contained in blocks A, B, and C in the heatmap in *Figure 2D*.

**Source data 2**. Results from GSEA comparing zebrafish embryos ectopically expressing SPOP-wt or -F133V.

**Figure supplement 1**. *Spop* knockdown in zebrafish using morpholinos results in a developmental phenotype.

*Figure 2. continued on next page*

*Figure 2. Continued*

**Figure supplement 2**. SPOP wt and F133V form heterodimers.

**Figure supplement 3**. Dendrogram of human primary prostate cancer cases based on BRCA1 knockdown genes (MSigDB: M2748).

**Figure supplement 4**. *SPOP* mutation and alterations in DNA repair genes in prostate cancer.

---

SPOP-F133V, the most commonly mutated residue in prostate cancer (*Figure 2D*). Consistent with recently reported proteomic data (*Theurillat et al., 2014*) and heterodimerization between mutant and wild-type SPOP in our models (*Figure 2—figure supplement 2*), transcriptional responses to SPOP-F133V compared with SPOP-wt and Spop morpholino showed a pattern consistent with dominant negative, selective loss of function; SPOP-F133V correlated with SPOP-wt for some gene sets (Cluster B) and correlated with Spop morpholino for others (Cluster A) (*Figure 2D*, *Figure 2—source data 1*). Gene set enrichment analyses (GSEA) revealed gene sets involving DNA repair impacted by modulating SPOP function (*Figure 2—source data 2*). Notably, the transcriptional response to F133V correlated highly with BRCA1 inactivation (*Figure 2E*). While SPOP has been previously proposed as involved in DNA damage and repair (DDR) signaling based on in vitro experiments (*Zhang et al., 2014a*), these results for the first time implicate a functional role for SPOP in DDR signaling in vivo and suggest this function is selectively impaired by prostate cancer-derived SPOP mutations. To test this hypothesis in human prostate cancers, we performed unsupervised hierarchical clustering of transcriptional data from 11 SPOP mutant and 53 SPOP wild-type tumors, based on the transcriptional signature of BRCA1 inactivation (MSigDB: M2748). We observed significant segregation of SPOP mutant tumors (p-value = $9.5e^{-07}$, *Figure 2F*) using this signature, with less robust segregation of tumors harboring CHD1 or MAP3K7 deletions co-occurring with SPOP mutation (*Figure 2—figure supplement 3*). These data suggest that human prostate cancers with *SPOP* mutations show transcriptional effects similar to BRCA1 inactivation, consistent with a role for SPOP in DSB repair. Supporting this hypothesis, we observed a trend of mutual exclusivity between *SPOP* mutations and *BRCA1* alterations in genomic data (p-value = 0.0496, *Figure 2—figure supplement 4*).

Consistent with the hypothesis that SPOP is involved in DSB repair in prostate cells, SPOP forms nuclear foci in prostate cells after γ-irradiation (*Figure 3A*) and stable expression of SPOP-wt conferred resistance to DSB-inducing agents cisplatin and camptothecin (CPT) (*Figure 3—figure supplement 1A,B*). To further define the impact of SPOP mutation in response to DNA damage, we utilized primary prostate cells isolated from transgenic mice with Cre-dependent conditional expression of SPOP-F133V. After transduction with Tamoxifen-inducible Cre (Cre-ERT2), cells were treated with 4-OH tamoxifen or vehicle and exposed to ionizing radiation (IR) (*Figure 3—figure supplement 2*). Spop localized in nuclear foci similar to human LNCaP cells in mouse prostate cells (MPCs) after IR, with no alteration in foci by expression of SPOP-F133V (*Figure 3A*, *Figure 3—figure supplement 1I*). Induction of SPOP-F133V resulted in delayed recovery from IR-induced damage as measured by comet assay (*Figure 3B*). No differences were seen in apoptosis after IR, as measured by PARP cleavage (*Figure 3—figure supplement 2*), subG1 events, or caspase-3 cleavage (data not shown). To functionally characterize how SPOP mutations affect response to DSB, we expressed SPOP-wt and SPOP-F133V in benign prostate epithelial cells (RWPE) and prostate cancer cells (22Rv1), and examined the induction, recognition, and resolution of CPT-induced DSBs. We found that DSB formation (measured by γH2AX foci and protein levels) was not affected by modulating SPOP function with siRNA, or expression of wildtype or mutant SPOP (*Figure 3C*, *Figure 3—figure supplement 1C,D*). Furthermore, there were no observed differences in early DNA damage signaling events, such as phosphorylation of ATM, ATR, Chk1, and Chk2 (*Figure 3—figure supplement 1E–H*), indicating that SPOP did not affect initial induction and recognition of DSBs or initial steps in DDR signaling. Consistent with this, SPOP showed only limited co-localization with early markers of DSB (γH2AX, phospho-ATM) in irradiated prostate cells (*Figure 3—figure supplement 1I*).

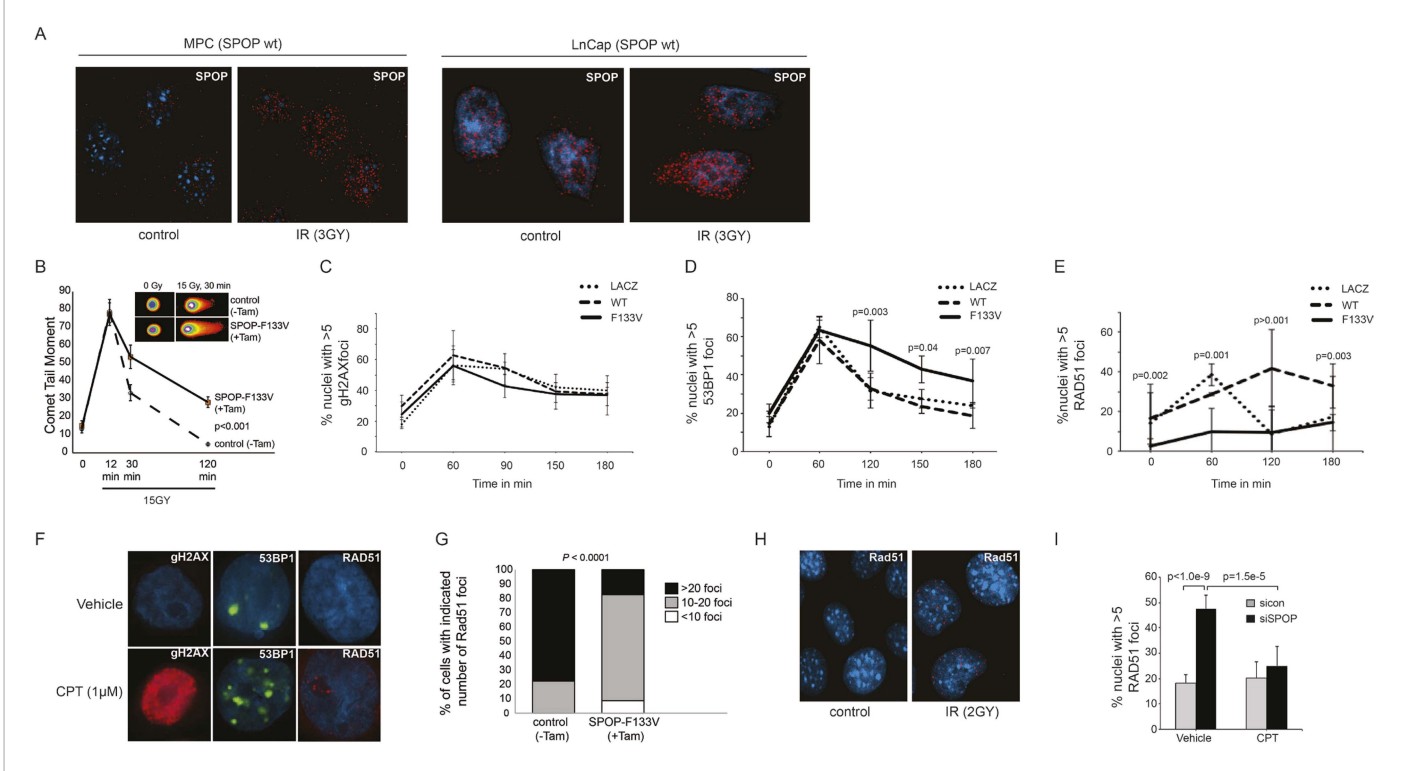

**Figure 3**. SPOP mutation impairs HDR and promotes NHEJ and SPOP-wt modulates DSB repair activity similar to BRCA1. (**A**) SPOP forms nuclear foci after induction of DNA damage by γ-irradiation (3GY) in prostate cells derived from transgenic mice (MPC) and human LNCaP cells. Red represents nuclear Spop protein foci. Blue represents nuclear DNA stained with DAPI. (**B**) MPC expressing Cre-inducible SPOP-F133V was infected with tamoxifen-inducible Cre (CreERT2), and DNA damage was assessed after IR with comet assays. Inset: representative cells showing comet tails after IR. (**C, D, E**) Quantification of γH2AX, 53BP1, or RAD51 foci in RWPE cells overexpressing WT or F133V mutant SPOP after camptothecin (CPT) (1 μM) induced DNA damage. Time indicates the observation interval in minutes including double strand break (DSB) induction (0–60 min) and recovery (60–180 min). Shown are the percentages of cells for each genotype with more than 5 foci per nucleus. Results are represented as s.e.m. (**F**) Representative pictures showing γH2AX, RAD51, or 53BP1 foci (red or green). Blue represents nuclear DNA stained with DAPI. (**G**) Quantification of Rad51 foci in γ-irradiated (2GY) MPC with tamoxifen-inducible SPOP-F133V. Rad51 foci were counted 30 min post irradiation. (**H**) Representative pictures showing Rad51 foci in mouse prostate epithelial cells before and after γ-irradiation (2GY). (**I**) Quantification of RAD51 foci in RWPE cells treated with siSPOP or control siRNA and subsequently exposed to CPT (1 μM, 1 hr).

The following figure supplements are available for figure 3:

**Figure supplement 1**. SPOP and response to DSB.

**Figure supplement 2**. Prostate cells derived from transgenic mice (MPC) expressing Cre-inducible SPOP-F133V were infected with tamoxifen-inducible Cre (CreERT2) and treated with 4-OH tamoxifen (+Tam) or vehicle (−Tam) and exposed to 15 GY IR followed by Immunoblot for SPOP, PARP, and vinculin (loading control) blot after 3 hr recovery.

**Figure supplement 3**. SPOP mutation impairs HDR-DSB repair and promotes NHEJ.

We next examined specific markers (53BP1, RAD51) of the two major DSB repair pathways, HDR and NHEJ. RAD51 is a component of the HDR pathway and a marker for engagement of the HDR machinery (*Baumann et al., 1996*). 53BP1 is a positive regulator of NHEJ that blocks 5′-DNA-end resection and therefore functions at the intersection of HDR and NHEJ; if 53BP1 is not cleared from sites of DSB by HDR components, it promotes error prone NHEJ (*Panier and Boulton, 2014*). Strikingly, SPOP-F133V-expressing prostate cells showed delayed clearance of 53BP1 from sites of DSB (*Figure 3D,F*, *Figure 3—figure supplement 3A*); similar effects were seen with another SPOP mutation (F102C) commonly observed in prostate cancer (*Figure 3—figure supplement 3I,J*). Furthermore, SPOP-wt increased RAD51 foci formation compared to controls, while SPOP-F133V-expressing cells showed a

dramatic decrease in RAD51 foci formation (*Figure 3E,F*, *Figure 3—figure supplement 3B*), consistent with impairment of HDR. Induction of SPOP-F133V in primary MPCs similarly decreased Rad51 foci after IR (*Figure 3G,H*). Knockdown of *SPOP* also resulted in a decrease of RAD51 foci, suggesting a selective loss of function of SPOP-F133V in HDR (*Figure 3I*). We also observed decreased clearance of 53BP1 foci and decreased RAD51 foci formation in SPOP-F133V-expressing cells after gamma irradiation, indicating that this effect is not specific to one mechanism of DSB induction (*Figure 3—figure supplement 3C–H*). The observed changes in DSB repair were not accompanied by changes in the cell cycle distribution of cells expressing SPOP-wt or F133V under these conditions (*Figure 3—figure supplement 3K*).

We next investigated the role of SPOP in DSB repair using the well established DR-GFP and Pem1-Ad2-EGFP reporter assays as functional readouts for HDR and NHEJ, respectively (*Figure 4A,D*) (*Pierce et al., 1999*; *Seluanov et al., 2004*). In the DR-GFP assay, knockdown of *SPOP* by siRNA decreased HDR competence in human epithelial cells to a similar level of *BRCA1* knockdown (*Figure 4B*). Conversely, ectopically expressed SPOP-wt increased the HDR competence in these cells, with partial loss of this function by mutant SPOP (*Figure 4C*). In contrast, the Pem1-Ad2-EGFP NHEJ reporter assay indicated an increase of NHEJ activity in SPOP-F133V expressing epithelial cells, while both *SPOP* siRNA and *BRCA1* siRNA increased NHEJ (*Figure 4E,F*). Taken together, these results suggest that SPOP promotes HDR, while somatic mutation in *SPOP*, as observed in prostate cancer with increased genomic rearrangements, impairs HDR and promotes error-prone NHEJ.

Human cancers with underlying defects in HDR (such as *BRCA1* inactivated breast and ovarian cancers) (*Bryant et al., 2005*; *Farmer et al., 2005*; *Audeh et al., 2010*; *Tutt et al., 2010*) show sensitivity to poly (ADP-ribose) polymerase 1 (PARP) inhibition. To test if SPOP inactivation conferred sensitivity to PARP inhibition after DSB induction, we utilized siRNA targeting SPOP in human prostate cancer cell lines (PC-3, LnCap, 22Rv1), followed by irradiation (5GY) and incubation with the PARP inhibitors olaparib or veliparib. Reduction of SPOP expression increased the sensitivity of these prostate cancer cells to both PARP inhibitors (*Figure 5A–C*, *Figure 5—figure supplement 1B,C*). To test if SPOP mutation also sensitizes to PARP inhibition, we treated MPCs (control vs tamoxifen-induced SPOP-F133V) with olaparib followed by IR—expression of SPOP-F133V increased sensitivity

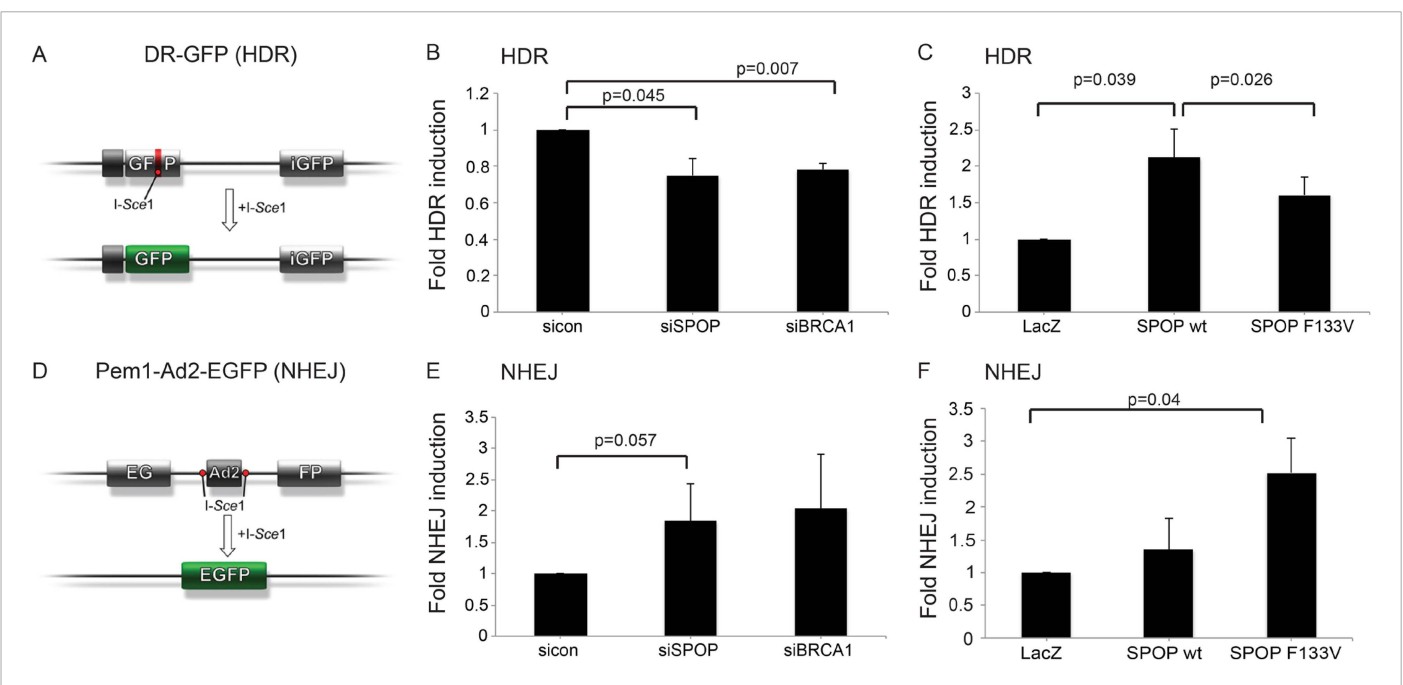

**Figure 4**. SPOP modulates DSB repair activity similar to BRCA1. (**A**) Schematic overview of the DR-GFP assay used to measure homology-directed repair (HDR) activity. (**B**, **C**) Analysis of the HDR-activity in HEK 293 cells with siRNA knockdown of SPOP or BRCA1 and ectopically expressing SPOP-wt or SPOP-F133V. (**D**) Schematic overview of the Pem1-Ad2-EGFP assay used to measure NHEJ-activity. (**E**, **F**) Analysis of the NHEJ-activity in HEK 293 cells with siRNA knockdown of SPOP or BRCA1 and ectopically expressing SPOP-wt or SPOP-F133V. All results are represented as s.e.m.

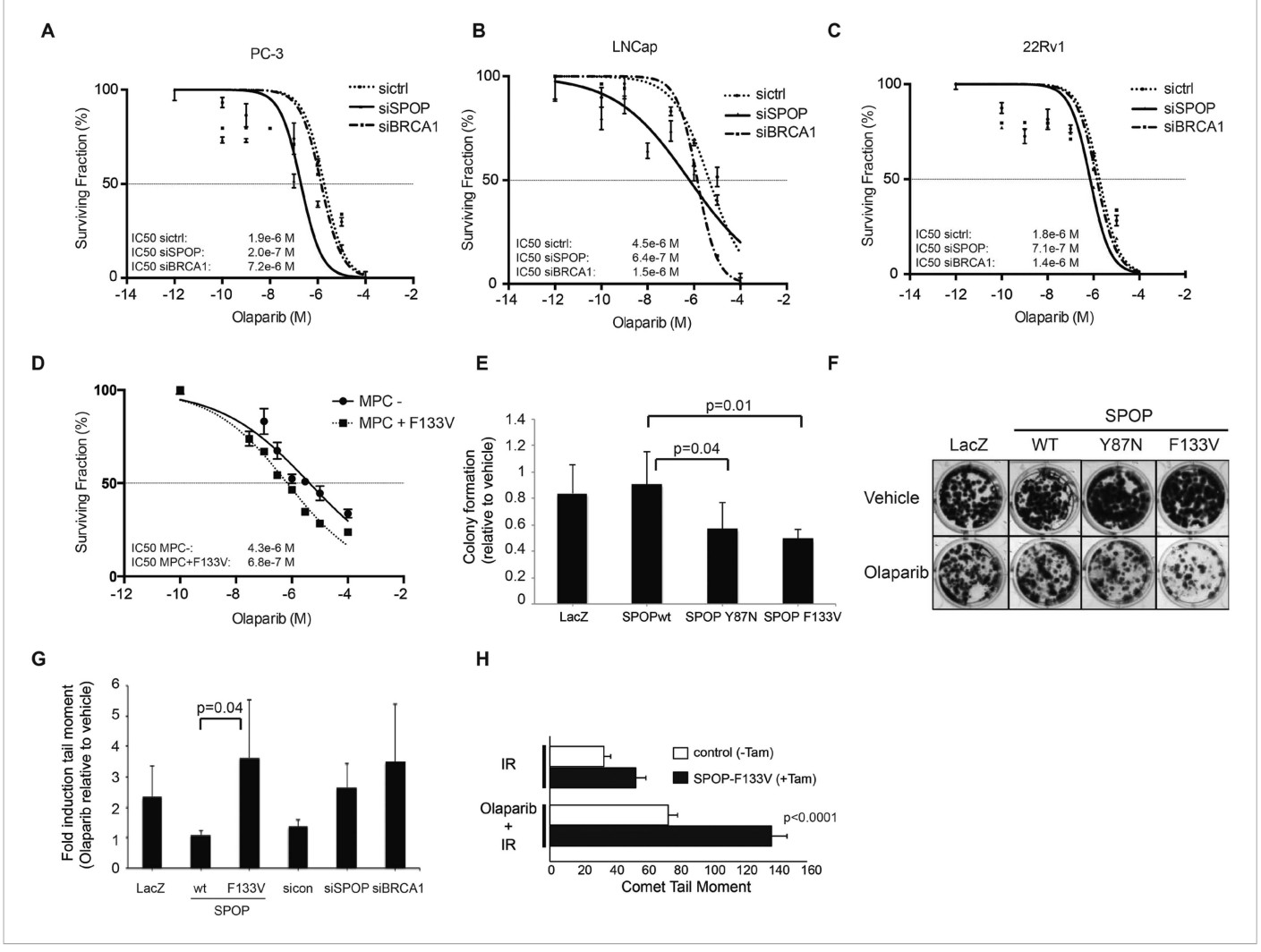

**Figure 5**. SPOP mutation sensitizes cells to therapeutic PARP inhibition. (**A–D**) Analysis of sensitivity to the PARP inhibitor olaparib in irradiated (5GY) prostate cancer (PC-3, LNCaP, 22Rv1) and mouse prostate epithelial cells (MPC) after *SPOP* knockdown (siSPOP) or tamoxifen-inducible SPOP[F133V] expression. *BRCA1* knockdown (siBRCA1) and non-targeting siRNA (sictrl) served as positive or negative control. The IC50 of olaparib for each genotype is indicated in Molar (M). (**E**) Analysis of the sensitivity of most frequently occurring prostate-specific SPOP mutants Y87N and F133V to olaparib in clonogenic assays using HEK293 cells. (**F**) Representative examples from the clonogenic assay used to assess long-term survival in HEK293 cells stably expressing SPOP-wt or SPOP mutants. (**G**) Impact of SPOP mutation on induction of genomic instability in 22Rv1 cells after olaparib (1 µM) treatment as measured by comet assay. Increased genomic instability was measured by an increase in the tail moment. (**H**) Genomic instability in tamoxifen-inducible SPOP-F133V expressing mouse prostate cells after γ-irradiation (15GY) with and without olaparib (1 µM) treatment as measured by comet assay. All results are represented as s.e.m.

The following figure supplement is available for figure 5:

**Figure supplement 1**. SPOP mutation and loss sensitize prostate cancer cells to therapeutic PARP inhibition.

to olaparib similar to loss of SPOP in human prostate cancer cell lines (*Figure 5D*). This was further confirmed in 22Rv1 cells ectopically expressing SPOP-F133V, which also showed an increased sensitivity to olaparib in viability assays, while cells ectopically expressing SPOP-wt were relatively resistant (*Figure 5—figure supplement 1A*). We further confirmed these results in HEK293 cells stably overexpressing the two common SPOP mutants (Y87N, F133V) in clonogenic survival assays (*Figure 5E,F*).

To determine if the altered sensitivity to PARP inhibitors was associated with increased DNA damage, consistent with impaired double-strand break repair, we performed single-cell gel

electrophoresis (comet assay) in prostate cells treated with olaparib. The comet assay revealed an increase of genomic instability in SPOP-F133V RWPE cells after PARP inhibition, similar to *SPOP* (and *BRCA1*) knockdown with siRNA (*Figure 5G*). Similarly, primary MPCs expressing SPOP-F133V showed increased damage after olaparib treatment (*Figure 5H*). Taken together, these data argue that SPOP mutation confers sensitivity to PARP inhibition due to impaired error-free HDR DSB repair and an increase in error-prone NHEJ.

## Discussion

In summary, here we report that SPOP, the substrate recognition component of an E3 ubiquitin ligase, is a regulator of the HDR-based DSB repair machinery. Using functional genetic approaches, we find that prostate-specific SPOP mutants deregulate DSB repair by promoting the error-prone NHEJ pathway (*Figure 6*). Importantly, after DSB induction, this loss of function leads to an increased sensitivity of mutant SPOP prostate cancer cells to PARP inhibition. These observations provide the first mechanism for the increased genomic instability of SPOP mutant prostate cancer, but also suggest that similar to other cancers with impaired HDR, this distinct class of cancer might benefit from treatment with clinically established DNA damaging therapeutics. This study therefore provides a rationale for hypothesis-based biomarker-driven clinical trials using PARP inhibitors or other DNA damaging agents in patients with prostate cancer.

   Genomic rearrangements represent critical events deregulating prostate cancer genomes and driving tumorigenesis, although among individual cancer samples, there is marked variability in number and character of structural rearrangements. Here, we identified a genomically unstable subclass of primary prostate cancer characterized by dramatically increased intrachromosomal rearrangements. This subset of clinically localized, treatment-naive prostate cancers display a degree of genomic instability previously thought only to occur in late stage, metastatic tumors. Importantly, increased rearrangements likely result in higher total SCNA burden, which is associated with more aggressive clinical behavior (*Hieronymus et al., 2014*; *Lalonde et al., 2014*).

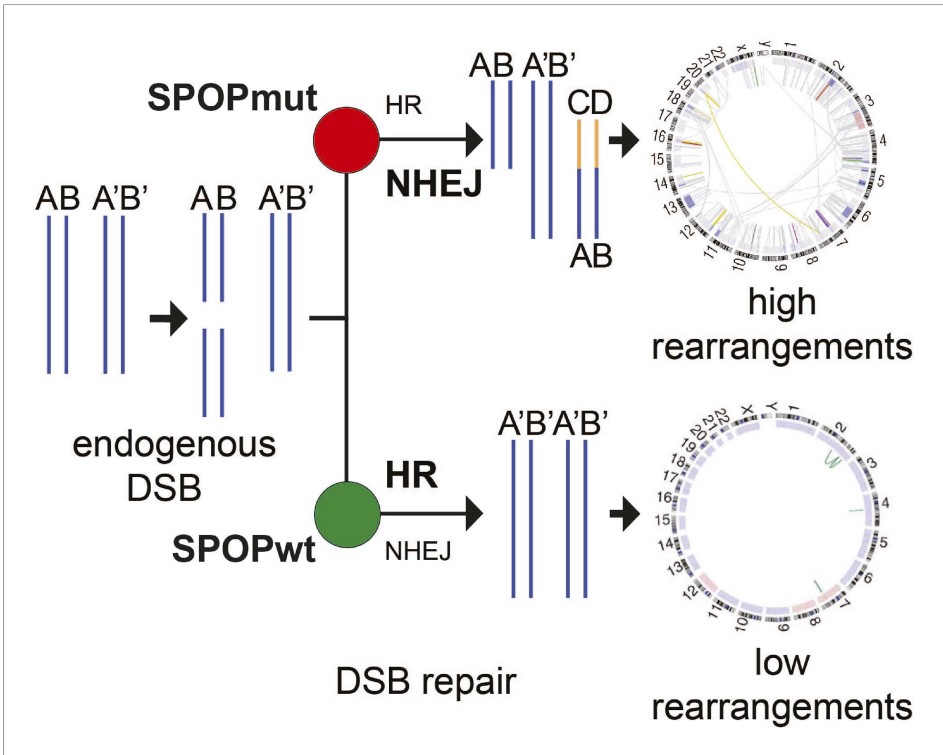

**Figure 6**. Proposed model of the effects of SPOP mutation on genome instability. In prostate epithelial cells, SPOP-wt promotes error-free HR and maintains genome stability. SPOP mutation impairs HR repair and promotes error-prone NHEJ, leading to increased genomic instability.

Increased rearrangements were associated with mutations in *SPOP*, encoding the substrate-recognition component of an E3 ubiquitin ligase complex. *SPOP* mutations are the most common point mutations in prostate cancer, but their role in promoting prostate cancer pathogenesis remains unclear. SPOP mutations occur early in the history of prostate cancer, based on clonality analysis reported here and presence in the prostate cancer precursor high grade prostatic intraepithelial neoplasia (HG-PIN) (*Barbieri et al., 2012*), potentially consistent with a 'gatekeeper' role in genome maintenance.

Consistent with increased intrachromosomal rearrangements in *SPOP* mutant cancers, in vivo data nominated a functional role for SPOP in DSB repair, similar to BRCA1. Interestingly, *SPOP* loss of function induced a developmental phenotype of neuronal degeneration and apoptosis; similar effects have been reported with other genes modulating DNA repair, including *BRCA1* (*Pulvers and Huttner, 2009*).

*SPOP* encodes the substrate recognition component of an E3 ubiquitin ligase, and here we identified a role for SPOP in regulating the HDR-based DSB repair machinery. Notably, many established components of the DNA damage response are components of enzymes regulating ubiquitylation (*Ceol et al., 2011*). A number of substrates have been reported as deregulated by prostate cancer-derived SPOP mutations, but the relevance of these in vitro findings to human prostate cancers is still unclear (*Geng et al., 2013, 2014*; *An et al., 2014*; *Theurillat et al., 2014*; *Zhang et al., 2014b*). In contrast, we focused on the phenotypes observed in human prostate cancers harboring *SPOP* mutations. Using functional genetic approaches, we find that prostate-specific SPOP mutants deregulate DSB repair by promoting the error-prone NHEJ pathway (*Figure 4F*).

The clinical importance of our findings is timely as potent PARP inhibitors, inducing synthetic lethality in cancers with alterations in DSB repair, are being evaluated and utilized in human cancers (*Fong et al., 2009*; *Tutt et al., 2010*; *Schiewer and Knudsen, 2014*). Here, we define a novel biomarker for increased genomic instability in clinically localized prostate cancer, provide the first mechanism by which SPOP mutations induce these alterations, and also suggest that similar to other cancers with impaired HDR, this distinct class of cancer may benefit from treatment with clinically established DNA damaging therapeutics, providing a rationale for genotype-based clinical trials.

## Materials and methods

### Human cell lines

Cells were purchased from ATCC and cultured according to the manufacturer's instructions. All cells were maintained with 1% penicillin/streptomycin (15140-122, Gibco, Grand Island, NY, United States). Cells were treated with camptothecin (C9911, Sigma-Aldrich, St. Louis, MO, United States), olaparib (AstraZeneca, United Kingdom), veliparib (Selleckchem, United Kingdom), or cisplatin (Sigma-Aldrich) at the indicated concentrations. For SPOP expression, lentiviral vectors (pLenti6/V5-Dest Gateway vector, V496-10, Invitrogen, Grand Island, NY, United States) coding for mutant or wild-type SPOP were used. Viruses were titrated to equalize infectivity and protein expression. Viral titers were measured by a p24 ELISA (Lenti-X p24 rapid titer kit, PT5002-2, Clontech, Mountain View, CA, United States). Viral transduction was performed by using Polybrene transfection reagent (TR-1003-G, Millipore, Billerica, MA, United States) at a final concentration of 8 ng/μl. Stable cell lines for WT or mutant SPOP were generated by pLenti viral transduction and subsequent selection by cell sorting for RFP using an Aria-II cell sorter (BD Biosciences, San Jose, CA, United States). Stable selection was monitored over time by confirming RFP- and MYC-tagged SPOP expression.

### Primary mouse prostate cells from conditional SPOP-F133V transgenic mice

#### Gene targeting

All mouse studies were approved by the WCMC IACUC under protocol 2012-0065. To generate mice conditionally expressing flag-myc tagged human *SPOP^{F133V}* pBTGinv (kind gift from Dr Yu Chen) was used to clone the cDNA subsequently to two inverted loxP sites, as previously described by *Chen et al. (2013)*. This vector, pBigT-invloxP-SPOP^{F133V}-IRES-nlsEFP was cloned into Rosa26-Pam1 (Addgene 15036). Vector was targeted into ES cells derived from albino C57B6 mice and injected into C57B6 blastocysts. Cre induction of SPOP-F133V was confirmed in ES cells. *R26^{F133V/+}* mice were generated and confirmed though standard genotyping.

## Primary mouse prostate cell isolation

Murine prostates were isolated from 10-week-old $R26^{F133V/+}$ mice and primary MPC lines were generated as described by Lukacs et al. (2010). Digested tissue was immediately mixed with growth factor reduced matrigel, plated in 36 drops and covered with organoid media as described by Karthaus et al. (2014), for 3–4 days. After 3–4 day of 3D culture cells were plated onto collagen-coated plates and grown as 2D cultures.

## Cre-induction and sorting

Stable MPC cells were transduced with a retrovirus expressing Cre-ERT2 (kind gift from Yu Chen) followed by puromycin selection. Following 48-hr treatment of the MPC cells with 1 µM 4-hydroxytamoxifen (Sigma-Aldrich) or vehicle, cells were sorted for GFP positivity using FACSAria II cell sorter (BD Biosciences). Vehicle-treated cells were sorted through a similar process.

## siRNA transfection

Reagents for siRNA knockdown of SPOP and BRCA1 were purchased from Dharmacon (Lafayette, CO, United States). The siGenome smartpools were used to target SPOP (M-017919-02-0005), BRCA1 (M-003461-02-0005) as well as non-targeting control (D-001206-13-05). The siRNA oligonucleotides were transfected with Lipofectamine 2000 at a final concentration of 50–100 pmol and incubated for 48 hr.

## Immunofluorescence

Cells were cultured on gelatin (0.1%) coated coverslips in 12-well plates, treated with CPT (1 µM, 1 hr) and then recovered for 2 hr in medium without CPT. Cells were collected at indicated time points and fixed in 4% PFA for 20 min, permeabilized in PBS with 0.5% Triton X 100 (EMD, TX1568-1) and blocked in 1% BSA in 1× PBS (P0195, Teknova, Hollister, CA, United States) for 30 min. Human prostate cells were then incubated in anti-Phospho (Ser139) H2AX (NBP1-64745, 1:500, Novus Biologicals, Littleton, CO, United States), anti-MYC-tag (clone 4A6, Millipore, 1:300), anti-RAD51 (PC130, 1:500, Calbiochem, United Kingdom), or anti-53BP1 (NB 100-904, Novus Biologicals, 1:500) primary antibodies followed by Alexa Fluor 555 or Alexa Fluor 488 labeled anti-mouse or anti-rabbit secondary antibodies (1:500) (A-21422, A-1101, Life Technologies). Mouse prostate epithelial cells were incubated in anti-phospho ATM (ab36810, 1:500, Abcam, United Kingdom), anti-Rad51 (Abcam, ab88572, 1:100), anti-γH2AX (Abcam, ab26350, 1:1000), and anti-SPOP (developed in the Rubin lab, 1:300). Pictures were taken on a Zeiss LSM 510 Laser Scanning Confocal Microscope.

## Immunoblot analysis

Cells were lysed as previously described (Barbieri et al., 2012). Proteins were quantified by BCA protein assay kit (23227, Thermo Scientific). 25–50 µg of total proteins were loaded onto a 4–15% gradient SDS-PAGE (Mini Protean TGX, 456-1084, Bio-Rad, Hercules, CA, United States) and subsequently transferred to PVDF membranes (Bio-Rad). Membranes were incubated with the following primary antibodies at the indicated dilutions: anti-SPOP rabbit monoclonal antibody (Epitomics), anti-MYC-tag (for SPOP) (Millipore, cat.no. 05-724, 1:1000), anti-phospho H2AX (pSer139) (NBP1-64745, Novus Biologicals, 1:1000), anti-total H2AX (NBP1-61896, Novus Biologicals, 1:1000), anti-RAD51 (ab133534, abcam, 1:1000), anti-BRCA1 (ab131360, abcam, 1:1000), anti-CHK1 (2G1D5, 1:1000, Cell Signaling, Danvers, MA, United States), anti phospho CHK1 (2348, Cell Signaling, 1:1000), anti-CHK2 (2662, Cell Signaling, 1:1000), anti-phospho CHK2 (2661, Cell Signaling, 1:1000), anti-ATR (Cell Signaling, 1:1000), anti-phospho ATR (2853, Cell Signaling, 1:1000), anti ATM (07-1286, Millipore, 1:1000), anti-phospho ATM (5883, Cell Signaling, 1:1000), anti-GAPDH (AB2302, Millipore, 1:5000). Primary antibodies were detected with HRP-conjugated anti-rabbit (32260, 1:10.000, Grand Island, NY, United States), anti-mouse (32230, Pierce, 1:10.000), anti-chicken (31401, Pierce, 1:5000) or HRP-conjugated streptavidin (18-152, Millipore, 1:5000).

## In vitro drug sensitivity assay

Human prostate and mouse prostate–epithelial cells (1000–5000) were seeded into 96-well microtiter plates (CLS3610-48EA, Sigma Aldrich). After irradiation (5GY) cells were treated with olaparib or veliparib in a concentration range of 0.0001 µM–100 µM for 72–96 hr. DMSO was used as control. Cell viability was measured by using the CyQuant NF cell proliferation assay (Life Technologies, C35006)

according to the manufacturer's instructions. Data normalization and nonlinear regression to calculate IC50 values was performed with Graph Pad Prism.

## Clonogenic survival assays

Stable SPOP-expressing HEK293 or 22Rv1 cells were treated with cisplatin (24 hr, concentration as indicated) or olaparib (1 µM, 4 days). 50 cells were seeded per well of a 12-well plate, grown for 10–12 days, fixed and stained with ethanol (10%) and crystal violet (0.09%). Colonies containing 50 cells or more were counted, and the drug cytotoxicity was calculated as the ratio of colonies formed after drug treatment relative to colonies from untreated cells.

## Single-cell gel electrophoresis

Single-cell gel electrophoresis (comet assay) was performed according to the manufacturer's instructions (Trevigen, Gaithersburg, MD, United States). $10^5$ cells/ml were mixed with molten LM agarose in a ratio of 1:10 and spread on a comet slide. Cells were subsequently lysed and subjected to electrophoresis. Cells were fixed and stained with SYBR Green. Images were collected with a 20× objective lens using a monochrome camera (CV-M4+CL, Jai, Japan) fitted in an Olympus BX51 microscope (Olympus, Center Valley, PA, United States). COMETscore v1.5 was used for image analysis, and 50–100 individual cells were analyzed per condition. The results are shown as comet tail moments, which is a representation of the fluorescence intensity in the tail relative to the head.

## DR-GFP (HDR) and Pem1-Ad2-EGFP (NHEJ) reporter assay

Cells were electroporated using the cell line nucleofactor kit V (VCA-1003, Lonza, Allendale, NJ, United States) with positive control plasmids (pNZE) and combinations of HDR-(pDRGFP) or NHEJ-(pPEM1-Ad2-EGFP) reporter constructs and an expression vector for the restriction enzyme I-SceI as described previously (*Pierce et al., 1999*). All the plasmids used in the GFP-reporter assay were a kind gift of Maria Jasin, Memorial Sloan Kettering Cancer Center, New York. The GFP expression induced by the positive control plasmid was used to normalize the electroporation efficiency. Negative controls contained the reporter construct and empty expression vectors lacking I-SceI. Cells were grown for 48 hr and processed for further flow cytometry analysis.

## Flow cytometry

Cells were fixed in PFA (2% vol/vol in PBS), permeabilized in 0.5% Triton X 100, and blocked in 1% BSA. Cells were incubated overnight in mouse Alexa Fluor 647 labeled anti-MYC-Tag antibodies (clone 9B11, cat.no: 2233, Cell Signaling, 1:50). SPOP (MYC-tag) and GFP expression was quantified by the LSRII flow cytometer (BD Biosciences).

## Cell cycle analysis

RWPE cells were transduced with LacZ, WT SPOP, or F133V SPOP lentiviral particles. Transduced cells were treated with Nocodazole (40 ng/ml) for 18 hr to synchronize cell cycle. Cell cycle distribution was determined by FACS 24 hr after culturing synchronized cells in normal growth media and labeling cells with propidium iodide (PI).

## Zebrafish

Wild-type zebrafish (strain AB/T) were maintained as described previously (*Ceol et al., 2011*). For functional studies, embryos were harvested at the one cell stage and yolk injected with SPOP-targeting morpholino (final 2–4 ng) or SPOP mRNA at the indicated concentrations. For phenotypical analysis, embryos were monitored for 5 days. RNA was isolated using Trizol reagent (15596-026, Life Technologies) at 24 hpf for RNA sequencing after dechorionating the embryos with pronase. For cell cycle analysis, embryos were homogenized and individual cells were stained with propidium iodide. All protocols were performed with prior approval of the WCMC IACUC (2011-0026).

## Zebrafish whole mount apoptosis assay

Embryos (24 hpf) were fixed in 4% PFA overnight. Embryos were than washed in PBS Tween (PBT) and Methanol, rehydrated in a series of Methanol-PBT dilutions (75% MeOH/25% PBT, 50% MeOH/50% PBT, 25% MeOH/75% PBT), and washed three times in PBT. Afterwards embryos were digested with PK (5 min, 10 µg/ml). PK digestion was followed by two PBT washes, a refixation in 4% PFA (20 min), and a postfixation in prechilled Ethanol:Acetic acid (2:1) (−20°C, 10 min). TUNEL staining was performed by using the ApopTag Peroxidase in situ apoptosis detection kit (S7100, Chemicon)

according to the manufacturer's instructions. TUNEL-positive cells were labeled by incubating the embryos in 3-3′-diaminobenzidine tetrahydrochloride (DAB).

## Morpholinos

Antisense oligos (morpholinos) targeting Spop were synthesized by Gene Tools (Corvalis, OR, United States). Morpholino 5 (MO5) was targeted to the sequence flanking the border between intron 1 and exon 2. Morpholino 7 (MO7) was targeted to the sequence flanking the border between intron 2 and exon 3. Morpholino sequences are shown in *Supplementary file 1*. Morpholino antisense oligos were dissolved in water to a final concentration of 1 mM. Primers used to validate morpholino-induced splicing of Spop are shown in *Supplementary file 1*.

## Plasmids

For the generation of pLenti SPOP expression vectors, mutant or SPOP WT was TOPO TA-cloned from pCMV-SPOP vectors (*Barbieri et al., 2012*) into pLenti viral vectors (pLenti6/V5-Dest Gateway Vector, V496-10, Invitrogen). For SPOP mRNA, in vitro transcription SPOP cDNA was TOPO TA cloned into a pCR 8 TOPO TA Gateway entry vector by using the pCR 8/GW/TOPO TA cloning kit (K2500-20, Invitrogen). 3-way-gateway recombination was then performed by using LR clonase II (11791, Invitrogen) to recombine SPOP cDNA from the pCR 8 TOPO TA gateway middle entry vector into a pDestTol2pA2 attR4-R3 (#394) destination vector. The 5′ entry clone was (p5E-CMV/SP6, #382) and the 3′ entry clone was (p3E-poly A, #302). The final vector contains a SP6 site for subsequent in vitro mRNA transcription.

## In vitro mRNA transcription

Human SPOP mRNA for rescue experiments in the zebrafish was generated using the mMessage mMachine SP6 kit (AM1340, Life Technologies) according to the manufacturer's instructions. RNA was analyzed quantitatively and qualitatively using the Agilent 2100 bioanalyzer.

## RNA sequencing

RNA was isolated from zebrafish embryos at 24 hpf using Trizol. 50 water- or morpholino-injected embryos were used. RNA libraries were created and sequenced using the GA2X or HighSeq genome analyzer (Illumina, United Kingdom) as described previously (*Barbieri et al., 2012*). The files first underwent quality control using FASTQC (v2_1.2.10). The vast majority of the reads from all the samples were of good quality (phred quality score >30). Therefore, no trimming or filtering of reads was required. The reads were then aligned with STAR (v2.3.0e) to zebrafish reference genome (Zv9).

## Bioinformatic analyses

Genomic analysis: SCNA profiles for 402 prostate cancers were obtained through processing of high-density oligonucleotide array data upon signal segmentation and then combined for genome-wide analysis for *SPOP* mutant and *SPOP* wild-type tumors. Clonality analysis for selected lesions was performed from sequencing data as in *Baca et al. (2013)*.

Differential Gene Expression (DGE) and Gene Set Enrichment Analysis: After sorting and indexing, the aligned files were used with bedtools (v2.16.1) and Ensembl Zebrafish Transcriptome (v70) to generate read counts for genes for all samples. These read counts were used with DESeq (2_1.2.10) for differential expression. Orthology was also added to the differential expression table in this step. Read counts were also counts per million (cpm) normalized using EdgeR (v3.4.2) package and then pre-ranked using Log2 Fold Change. GSEA (v2-2.0.13) was run in pre-ranked mode to identify enriched signatures. Primers used for validation of the expression of selected genes are shown in *Supplementary file 1*.

## Human samples

Clinically localized prostate cancers were selected for transcriptome sequencing as described (*Barbieri et al., 2012*). All samples were collected with informed consent of the patients and prior approval of the institutional review boards (IRB) of respective institutions. Additionally, the sequencing and data release of all transcriptome-sequenced samples was reviewed and approved by local IRB.

## Statistics

The results are represented as means of at least three experiments (standard error of the mean (s.e.m.) are indicated by error bars). Student's *t*-test was used to calculate the statistical significance of differences in the HR- and NHEJ-reporter assays. Fisher's exact test (two sided) was used to assess the

statistical significance of differences in the foci assays. Nonlinear regression was performed to determine the IC50 values in the in vitro drug sensitivity assays.

## Accession codes

All RNA-seq data are deposited in the NCBI sequencing read archive (SRA) under the accession number SRP063952.

## Acknowledgements

We are grateful to the individuals with prostate cancer and their families for contributing to this study. We thank The Cancer Genome Atlas Research Network (TCGA) for providing the prostate cancer genomic data. We thank M Jasin for providing the GFP-based HDR- and NHEJ- reporter constructs as well as critical reading and discussion of the manuscript. We thank F Vanoli and S Aziz for providing assistance with the establishment of the GFP-reporter and in vitro drug response assays. We thank F Feng and S Han for helpful discussions on assay design. We thank C Bourque for assisting with the zebrafish experiments.

This work was supported by the following grants: US National Cancer Institute (2R01CA125612-05A1, MAR, PZ, FD, and K08CA187417-01, CEB), AIRC IG13562 (FD), Prostate Cancer Foundation Challenge Award (MAR, PZ), Prostate Cancer Foundation Young Investigator Award (CEB), Urology Care Foundation Research Scholar Award (CEB), The Frederick J and Theresa Dow Wallace Fund of the New York Community Trust (CEB), Marie Curie International Incoming Fellowship (GB), Prostate Cancer UK project grant (JDB, GB), NIHR RM/ICR Biomedical Research Centre flagship grant (JDB, GB) and a seed grant from the Department of Medicine of New York Presbyterian Hospital (YH).

## Additional information

### Competing interests

CEB: A patent (US Patent Application No: 2013/0331,279) has been issued to Weill Medical College of Cornell University on SPOP mutations in prostate cancer; is listed as co-inventor. MAR: A patent (US Patent Application No: 2013/0331,279) has been issued to Weill Medical College of Cornell University on SPOP mutations in prostate cancer; is listed as co-inventor. The other authors declare that no competing interests exist.

### Funding

| Funder | Grant reference | Author |
| --- | --- | --- |
| National Cancer Institute (NCI) | 2R01CA125612-05A1 | Pengbo Zhou, Francesca Demichelis, Mark A Rubin |
| Prostate Cancer Foundation (PCF) | Challenge Award | Pengbo Zhou, Mark A Rubin |
| Urology Care Foundation (Urology Care Foundation, Inc.) | Research Scholar Award | Christopher E Barbieri |
| New York Community Trust (NYCT) | The Frederick J. and Theresa Dow Wallace Fund | Christopher E Barbieri |
| European Research Council (ERC) | Marie Curie Fellowship | Gunther Boysen |
| Prostate Cancer UK | PG13-036 | Gunther Boysen, Johann S de Bono |
| National Cancer Institute (NCI) | K08CA187417-01 | Christopher E Barbieri |
| Associazione Italiana per la Ricerca sul Cancro (AIRC) | IG13562 | Francesca Demichelis |
| Prostate Cancer Foundation (PCF) | Young Investigator Award | Christopher E Barbieri |

The funders had no role in study design, data collection and interpretation, or the decision to submit the work for publication.

## Author contributions

GB, CEB, DP, MB, S-SC, FD, Final approval of the version to be published, Conception and design, Acquisition of data, Analysis and interpretation of data, Drafting or revising the article; AD, SN, DH, CM, LX, JH, PL, SC, DL, Final approval of the version to be published, Acquisition of data, Analysis and interpretation of data, Drafting or revising the article; PZ, AS, JSB, YH, Final approval of the version to be published, Conception and design, Analysis and interpretation of data, Drafting or revising the article; MAR, Final approval of the version to be published, Conception and design, Analysis and interpretation of data, Drafting or revising the article, Contributed unpublished essential data or reagents

## Ethics

Animal experimentation: All protocols were performed with prior approval of the WCMC IACUC under protocol 2012-0065.

# Additional files

## Supplementary file

• Supplementary file 1. Sequences of antisense-morpholino's targeting Spop as well as primers used in this study.

## Major dataset

The following previously published dataset was used:

| Author(s) | Year | Dataset title | Dataset ID and/or URL | Database, license, and accessibility information |
|---|---|---|---|---|
|  | 2012 | Prostate Cancer Genome Sequencing Project | http://www.ncbi.nlm.nih.gov/ projects/gap/cgi-bin/study.cgi? study_id=phs000447.v1.p1 | Publicly available at the NCBI (Accession no: phs000447.v1.p1). |

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
