## [Decision Letter]

Thank you for submitting your work entitled “*SPOP* mutation leads to genomic instability in prostate cancer” for peer review at *eLife*. Your submission has been favorably evaluated by Charles Sawyers (Senior Editor) and three reviewers, one of whom is a member of our Board of Reviewing Editors.

The reviewers have discussed the reviews with one another and the Reviewing Editor has drafted this decision to help you prepare a revised submission.

The reviewers considered this manuscript very well written, interesting, and potentially very important for a number of reasons. The manuscript describes the characterization of *SPOP* mutations in prostate cancers and reveals a new role for SPOP in the DNA damage repair response (DDR). While analyzing data on chromosomal rearrangements and mutational status of hundreds of prostate cancers, the authors make the interesting observation that ‘high rearrangement’ correlates with *SPOP* mutations. They go on to show that SPOP is involved in late steps of the DDR response, and that *SPOP* mutations impair homologous recombination (HR) and favor error-prone non-homologous end-joining (NEHJ), which could potentially explain the correlation between *SPOP* mutations and high frequency of rearrangements. This function resembles that of the DDR factor BRCA1. In fact, *SPOP* mutations also seem to be mutually exclusive with *BRCA1* mutations. Finally, the authors show some data suggesting that *SPOP* mutations may sensitize cells to PARP inhibitors, which have been shown to be synthetic lethal with *BRCA1* mutations.

The reviewers considered the manuscript to be potentially meritorious for publication in *eLife* after addressing some major concerns:

1) The data supporting the conclusion that *SPOP* mutations sensitize cancer cells to PARP inhibitors is currently very weak, based solely on the analysis of RWPE primary prostate cells, RR2v1 prostate cancer cells and embryonic kidney cells (HEK) using artifact-prone overexpression experiments. The conclusion that *SPOP* mutations could be synthetic lethal with PARP inhibition could have profound implications in the clinic and therefore must be exhaustively tested by the authors. Toward this end, the following experiments are requested:

1.1) PARP inhibitors should be tested in a dose-response experiment in prostate cancer cell lines expressing wild type or mutant SPOP. We recognize that this request may be challenging due to the limited number of prostate cancer cell lines but we would like to see if the prediction holds in whatever models may be available.

1.2) The authors should establish an isogenic system using a prostate cancer cell line of choice currently expressing wild type SPOP and thus resistant to PARP inhibitors, and either knockdown or knock-out SPOP and demonstrate that SPOP depletion is sufficient to confer sensitivity.

2) Can the authors account for the discrepancy between their data and the previously published work by Zhang et al. (Carcinogenesis 35: 1691, 2014)? This paper shows that SPOP is required for the DDR. Accordingly, Boysen et al. should also test whether SPOP localizes to DSBs following IR and they should more thoroughly present their own DDR data, even though their results were negative.

3) The manuscript begins with examination of WGS data from 55 treatment naïve prostate cancers and subsequent distribution into 2 chromosomal rearrangement populations – “low-rearrangement” and “high-rearrangement”. The “high-rearrangement” group was significantly associated with *SPOP* mutations, but also with recurrent deletions on 5q and 6q, containing the tumor suppressors CHD1 and MAP3K7. Because of clonality studies the authors pursue the role of *SPOP* mutations only. Though the data regarding contribution of *SPOP* mutation to DNA damage response and sensitivity to PARP inhibitors are convincing, no attempt is made to see whether losses at 5q and 6q contribute to this phenotype independently or cooperatively with *SPOP* mutation. 5q CNAs, specifically loss of CHD1, have been implicated in high chromosomal rearrangements, and MAP3K7 is also highly correlated with chromosomal rearrangements. Examining the 5q and 6q status of the patients in Figure 2, either alone or in combination with *SPOP* mutant status would be informative and provide a clearer understanding of the contributions of these genes to the DNA damage phenotype (*BRCA1* mutant gene set) of this subtype. It would be informative to see how these CNAs co-segregate.

4) The paper does not provide a molecular mechanism. Specifically, recent studies by Theurillat et al. indicate that mutations in *SPOP*, an E3 substrate binding protein, results in upregulation of specific protein substrates (DEK, TRIM24, for example) via a dominant negative mechanism. But this new study by Boyser et al. does not provide a mechanism of disrupted HR. Is SPOP inhibiting the degradation of some critical substrate which is causing the transcriptional response consistent with *BRCA1* inactivation? Can the authors provide any additional data in this regard?

Minor comments [abridged]:

1) The authors conclude that phosphorylation of ATR is not affected by *SPOP* mutations, but the blots in Figure 3—figure supplement 1 are discordant in this sense. Please repeat with several biological replicates to define this issue.

2) In the zebrafish experiments, SPOP knockdown leads to cell death concurrently with induction of p53 mRNA and induced expression of the p53 target genes *MDM2* and *CCNG1*. The authors should comment on this, acknowledging that cell death could be due to p53 activation in this setting.

3) A recent study indicates that a high percentage (15-20%) of metastatic CRPC have underlying mutations in DNA repair genes (*BRCA2* and *ATM* most commonly). The authors show some TGCA data for *BRCA1*. Are these mutations in CRPC (i.e. *BRCA2* and *ATM*) mutually exclusive with *SPOP* mutations?

---

## [Author Response]

*1) The data supporting the conclusion that* SPOP *mutations sensitize cancer cells to PARP inhibitors is currently very weak, based solely on the analysis of RWPE primary prostate cells, 22Rv1 prostate cancer cells and embryonic kidney cells (HEK) using artifact-prone overexpression experiments. The conclusion that* SPOP *mutations could be synthetic lethal with PARP inhibition could have profound implications in the clinic and therefore must be exhaustively tested by the authors. Toward this end, the following experiments are requested*:

*1.1) PARP inhibitors should be tested in a dose-response experiment in prostate cancer cell lines expressing wild type or mutant SPOP. We recognize that this request may be challenging due to the limited number of prostate cancer cell lines but we would like to see if the prediction holds in whatever models may be available*.

We concur that the implications of *SPOP* mutant cancers as a target for PARP inhibition is potentially important. We initially sought prostate cancer cell line models that harbor the *SPOP* mutation but unfortunately there are none to our knowledge. We also recognize the limitations of overexpressing mutant SPOP in the 22Rv1 prostate cancer cell lines as a model system. Therefore to address this issue, we developed a model system expressing inducible mutant SPOP in mouse prostate organoids. These are derived from a genetically engineered mouse model expressing SPOP-F133V in a Cre dependent manner, using a knock-in to the *Rosa26* locus (manuscript in preparation). The protein expression levels of mutant SPOP are approximately the same as endogenous SPOP in this model, mimicking human tumors (Figure 3—figure supplement 2), and originally utilizing this model (Figure 3, Figure 5). We have expanded the series of experiments examining PARP inhibitor sensitivity in this mouse organoid model. In new dose response experiments suggested by the reviewer, expression of physiologically relevant levels of mutant SPOP led to increased sensitivity to olaparib compared to control cells (Figure 5).

*1.2) The authors should establish an isogenic system using a prostate cancer cell line of choice currently expressing wild type SPOP and thus resistant to PARP inhibitors, and either knockdown or knock-out SPOP and demonstrate that SPOP depletion is sufficient to confer sensitivity*.

In the initial submission, we presented limited data showing that knockdown of *SPOP*, similar to knockdown of *BRCA1*, sensitized 22Rv1 prostate cancer cells to PARP inhibition (Figure 5). We have now expanded the series of experiments examining PARP inhibitor sensitivity with *SPOP* knockdown in our model systems. *SPOP* knockdown resulted in increased sensitivity to PARP inhibition in three prostate cell lines (22Rv1, PC3, and LNCaP) (Figure 5). This effect was observed using two PARP inhibitors (i.e. olaparib, veliparib).

While these experiments do support the proposed role of SPOP in DSB repair, they should be interpreted with caution, since deletion of *SPOP* is not observed in human prostate cancer, and therefore may not phenocopy *SPOP* mutation in all models.

*2) Can the authors account for the discrepancy between their data and the previously published work by Zhang et al. (Carcinogenesis 35: 1691, 2014)? This paper shows that SPOP is required for the DDR. Accordingly, Boysen et al. should also test whether SPOP localizes to DSBs following IR and they should more thoroughly present their own DDR data, even though their results were negative*.

Zhang D. et al. proposed a role for SPOP in the DDR based primarily on co-localization with other DDR components, without focusing on cancer-derived *SPOP* mutations. Their paper showed that *SPOP* knockdown with siRNA potentially led to prolongation of DSB repair (as measured by γH2AX foci). While these studies examining the consequences of *SPOP* knockdown in other cell types are interesting, and suggest a role in the DDR, they were not examined in a prostate context, and *SPOP* mutations are highly specific to prostate cancer. The tissue context may account for some of the differences between our results. We agree with the reviewers that determining localization of SPOP following γ-irradiation (IR) and co-localization with DNA repair complexes in a prostate context is important. Using immunofluorescence, we show that SPOP has increased nuclear foci following IR in both mouse and human prostate cells (Figure 3).

In addition, we show that SPOP has only limited co-localization with γH2AX and phospho-ATM, following IR in prostate cells – this co-localization was not affected by expression of mutant SPOP (Figure 3—figure supplement 1).

These additional experiments support that SPOP plays a role in responding to DSB. However, the differences in co-localization compared to those reported by Zhang et al. may be due to tissue context. Further mechanistic experiments will be necessary (and are ongoing) to define the role of SPOP and impact of prostate cancer derived *SPOP* mutations.

*3) The manuscript begins with examination of WGS data from 55 treatment naïve prostate cancers and subsequent distribution into 2 chromosomal rearrangement populations – “low-rearrangement” and “high-rearrangement”. The “high-rearrangement” group was significantly associated with* SPOP *mutations, but also with recurrent deletions on 5q and 6q, containing the tumor suppressors CHD1 and MAP3K7. Because of clonality studies the authors pursue the role of* SPOP *mutations only. Though the data regarding contribution of* SPOP *mutation to DNA damage response and sensitivity to PARP inhibitors are convincing, no attempt is made to see whether losses at 5q and 6q contribute to this phenotype independently or cooperatively with* SPOP *mutation. 5q CNAs, specifically loss of CHD1, have been implicated in high chromosomal rearrangements, and MAP3K7 is also highly correlated with chromosomal rearrangements. Examining the 5q and 6q status of the patients in*
Figure 2*, either alone or in combination with* SPOP *mutant status would be informative and provide a clearer understanding of the contributions of these genes to the DNA damage phenotype (*BRCA1 *mutant gene set) of this subtype. It would be informative to see how these CNAs co-segregate.*

*SPOP* mutations, and deletions of 5q21 (CHD1) and 6q (MAP3K7) are highly associated, and together constitute a distinct molecular class of prostate cancer. The observation that all of these alterations were associated with increased genomic rearrangements suggests that this represents a genomically unstable molecular class of prostate cancer. To prioritize alterations for functional study, we used two points of rationale:

A) Because the phenotype in question is structural genomic rearrangements, genomic deletions (such as CHD1 and MAP3K7) could be causative or consequences of this phenotype, since they result from structural rearrangements. Point mutations (such as *SPOP*) cannot be a consequence of deregulated genomic rearrangements.

B) We therefore examined clonality data to determine the earliest event in this class (and therefore possibly the initiator of genomic instability, leading to subsequent genomic deletions), which prioritized *SPOP* mutation for functional study.

We appreciate the reviewers’ comment and understand that a better exposition of these ideas will improve the message of the manuscript. We have therefore added the following computational analyses and annotation:

A) To address the reviewers’ comments, we have now extended the genomic analysis to include the official TCGA-PRAD (prostate adenocarcinoma) dataset of 333 patients. The revised data for the copy number analysis include a total of 430 patients of which 47 harbor a *SPOP* mutation. As originally reported, deletions on 5q21.1 and 6q15, spanning *CHD1* and *MAP3K7* respectively, are significantly concurrent with mutations in *SPOP* (Figure 1—figure supplement 3). When comparing individually *CHD1* deletions, *MAP3K7* deletions, and *SPOP* mutations, we did not observe significant differences in genomic burden for any one lesion (Figure 1—figure supplement 4).

B) Figure panel 1F has now been updated with this larger set. The clonality fraction for *SPOP* mutation, *CHD1* deletion and *MAP3K7* deletion are 93.2%, 66.3%, and 88.1%, respectively.

C) Further, we studied the dependencies of the 3 lesions using the larger dataset. The data still supports *SPOP* mutations preceding CHD1 (Figure 1—figure supplement 5). These observations are based on algorithm we developed to build tumor evolution paths (3; 26).

D) To specifically query the contribution of *CHD1* and *MAP3K7* deletions to the DNA damage phenotype, we included the deletion status in the annotation panel of the dendrogram from Figure 2. The cluster enrichment for *CHD1* deletion is borderline significant (in line with the strong association with *SPOP* mutations); no signal is detected for *MAP3K7*. We favor the inclusion of this panel as Figure 2—figure supplement 3 while maintaining *SPOP* annotation only in Figure 2 as it is the focus of the study.

*4) The paper does not provide a molecular mechanism. Specifically, recent studies by Theurillat et al*. *indicate that mutations in* SPOP*, an E3 substrate binding protein, results in upregulation of specific protein substrates (DEK, TRIM24, for example) via a dominant negative mechanism. But this new study by Boyser et al. does not provide a mechanism of disrupted HR. Is SPOP inhibiting the degradation of some critical substrate which is causing the transcriptional response consistent with* BRCA1 *inactivation? Can the authors provide any additional data in this regard?*

In this current manuscript we focus on the biologic implications of *SPOP* mutations in the prostate cancer context. We agree wholeheartedly that the biochemical mechanism by which SPOP regulates DNA repair is a critical question for the field going forward. Defining the specific substrates potentially responsible for this biology will be a long process. For instance, BRCA1 is also a ubiquitin ligase clearly critical for DNA repair, yet the specific substrate of BRCA1 responsible for this function remains poorly defined over a decade after its discovery. This goal will require long-term mechanistic efforts outside the scope of the current manuscript.

Minor comments [abridged]:

*1) The authors conclude that phosphorylation of ATR is not affected by SPOP mutations, but the blots in*
Figure 3—figure supplement 1
*are discordant in this sense. Please repeat with several biological replicates to define this issue*.

The results presented originally were representative of at least three biological replicates (Figure 7). We found ATR phosphorylation in this experiment to be highly variable, especially in the 22Rv1 cell line – however, there was no consistent effect of SPOP siRNA or ectopic expression of wt or mutant SPOP on ATR phosphorylation.

Author response image 1.**DOI:**
http://dx.doi.org/10.7554/eLife.09207.027

*2) In the zebrafish experiments, SPOP knockdown leads to cell death concurrently with induction of p53 mRNA and induced expression of the p53 target genes* MDM2 *and* CCNG1*. The authors should comment on this, acknowledging that cell death could be due to p53 activation in this setting*.

The reviewers correctly note that SPOP knockdown triggered a transcriptional response similar to p53 activation. The developmental phenotype resulting from this knockdown was observed with two different SPOP morpholinos and was rescued with human wildtype SPOP mRNA suggesting a SPOP-specific effect. Interestingly, mutations in *SPOP* and p53 are mutual exclusive in localized prostate cancer (4), further supporting a potential relationship. We agree with the reviewers, and have acknowledged the p53 activation in the text of the results section.

*3) A recent study indicates that a high percentage (15-20%) of metastatic CRPC have underlying mutations in DNA repair genes (*BRCA2 *and* ATM *most commonly). The authors show some TGCA data for* BRCA1*. Are these mutations in CRPC (i.e.* BRCA2 *and* ATM*) mutually exclusive with* SPOP *mutations?*

The reviewers pose an excellent question – mutual exclusivity between *SPOP* mutations and *BRCA2* and *ATM* mutations described in the SU2C/PCF cohort (Robinson et al., Cell 2015) of CRPC samples would support a continued role of SPOP mutation in compromised DNA repair in CRPC.

As shown below (Figure 8), no CRPC samples harbor both *SPOP* mutation and *ATM* mutations. Two cancers showed both *SPOP* mutation and *BRCA2* mutation, and one sample had both an *ATM* mutation and *BRCA2* mutation. The sample size (n=150 total) and event rates preclude meaningful statistical analysis (all p-values non significant for analysis (all p-values non significant for analysis mutual exclusivity). We have added this analysis to Figure 2—figure supplement 4.

Author response image 2.**DOI:**
http://dx.doi.org/10.7554/eLife.09207.028